# Hypomethylation mediates genetic association with the major histocompatibility complex genes in Sjögren's syndrome

**Calvin Chi**[1,2], **Kimberly E. Taylor**[3], **Hong Quach**[2], **Diana Quach**[2], **Lindsey A. Criswell**[3⊚]*, **Lisa F. Barcellos**[1,2⊚]*

**1** Center for Computational Biology, College of Engineering, University of California, Berkeley, Berkeley, California, United States of America, **2** Genetic Epidemiology and Genomics Laboratory, School of Public Health, University of California, Berkeley, Berkeley, California, United States of America, **3** Department of Medicine, Russell/Engleman Rheumatology Research Center, University of California, San Francisco, San Francisco, California, United States of America

⊚ These authors contributed equally to this work.
* lbarcellos@berkeley.edu (LFB); lindsey.criswell@ucsf.edu (LAC)

## Abstract

Differential methylation of immune genes has been a consistent theme observed in Sjögren's syndrome (SS) in CD4+ T cells, CD19+ B cells, whole blood, and labial salivary glands (LSGs). Multiple studies have found associations supporting genetic control of DNA methylation in SS, which in the absence of reverse causation, has positive implications for the potential of epigenetic therapy. However, a formal study of the causal relationship between genetic variation, DNA methylation, and disease status is lacking. We performed a causal mediation analysis of DNA methylation as a mediator of nearby genetic association with SS using LSGs and genotype data collected from 131 female members of the Sjögren's International Collaborative Clinical Alliance registry, comprising of 64 SS cases and 67 non-cases. *Bumphunter* was used to first identify differentially-methylated regions (DMRs), then the causal inference test (CIT) was applied to identify DMRs mediating the association of nearby methylation quantitative trait loci (MeQTL) with SS. *Bumphunter* discovered 215 DMRs, with the majority located in the major histocompatibility complex (MHC) on chromosome 6p21.3. Consistent with previous findings, regions hypomethylated in SS cases were enriched for gene sets associated with immune processes. Using the CIT, we observed a total of 19 DMR-MeQTL pairs that exhibited strong evidence for a causal mediation relationship. Close to half of these DMRs reside in the MHC and their corresponding meQTLs are in the region spanning the *HLA-DQA1*, *HLA-DQB1*, and *HLA-DQA2* loci. The risk of SS conferred by these corresponding MeQTLs in the MHC was further substantiated by previous genome-wide association study results, with modest evidence for independent effects. By validating the presence of causal mediation, our findings suggest both genetic and epigenetic factors contribute to disease susceptibility, and inform the development of targeted epigenetic modification as a therapeutic approach for SS.

**Data Availability Statement:** Genotype data has been deposited in dbGaP (accession number phs000672.v1.p1). The metadata, non-normalized

data, and processed data of DNA methylation has been uploaded to GEO (accession number GSE166373).

**Funding:** L.C. was supported by the SICCA grant HHSN268201300057C (https://www.nidcr.nih.gov/), R03 grant R03DE024316 (https://grants.nih.gov/), and Sjögren's Syndrome Foundation grant (https://www.sjogrens.org/). C.C. was supported by the National Science Foundation Graduate Research Fellowship Program under Grant No. DGE 1106400 (https://www.nsf.gov). The funders had no role in study design, data collection and analysis, decision to publish, or preparation of the manuscript.

**Competing interests:** The authors have declared that no competing interests exist.

## Introduction

Sjögren's syndrome (SS) is an autoimmune disease characterized by the lymphocytic infiltration of salivary and lacrimal glands, resulting in dryness of the mouth and eyes, fatigue, and joint pain. The prevalence of SS is estimated to be 3% in individuals aged 50 years or older and 0.6% overall, with a 9:1 female-to-male predominance [1]. When SS occurs in isolation, it is referred to as primary SS; secondary SS co-occurs with other systemic autoimmune diseases [2]. Environmental factors including infectious agents, stress, air pollution, and silicone are implicated in disease pathogenesis [3–6]. Genetic association studies have established genetic loci both within and outside the major histocompatibility complex (MHC) [7–9].

Differential methylation has been reported by multiple studies of CD4+ T cells, CD19+ B cells, whole blood, and labial salivary glands (LSGs) in SS [10–21]. Specifically, hypomethylation of immune-related genes has been observed, along with implications for altered gene expression. Some of these studies found evidence supporting genetic control of DNA methylation. Miceli-Richard *et al.* reported an overlap of differentially methylated probes with established genetic risk loci, suggesting both genetic and epigenetic abnormalities in the same genes [18]. Imgenberg-Kreuz *et al.* identified methylation quantitative trait loci (meQTL), or loci where genetic variation is associated with DNA methylation, in whole blood [19]. However, this association analysis was performed based on the whole blood of healthy controls only, instead of based on both pSS cases and controls. These association results alone are not sufficient to support the causal mediation of DNA methylation for the genetic association with SS (e.g. ruling out reverse causation). Distinguishing differential methylation that is a cause of, rather than a consequence of, disease is essential for further consideration of epigenetic modification as a therapeutic approach to SS [22].

We investigated evidence for genetic control of DNA methylation for SS risk using LSGs from 64 primary SS cases and 67 symptomatic non-cases from the Sjögren's International Collaborative Clinical Alliance (SICCA) registry. Our overall approach first used *bumphunter* to identify differentially-methylated regions (DMRs), or regions where contiguous CpG sites are differentially methylated in the same direction. Then, for each DMR, we identified its corresponding meQTLs as SNPs within ±250 kb that are associated with its DNA methylation levels. These meQTLs are considered *cis*-meQTLs since meQTL effects spanning multiple megabase pairs at the MHC have been observed [23]. Finally, we performed the causal inference test (CIT) developed by Millstein *et al.* to find DMR-meQTL pairs where the DMR shows strong evidence of mediating the risk of surrounding meQTLs on SS [24]. By extension, this also suggested CpG sites whose methylation levels could be *independent* of neighboring genetic variation and CpG sites whose methylation levels may be influenced by disease status. These findings significantly expand what is known about potential targets of epigenetic-modifying agents within the human genome. Although cancer has been the most common application for epigenetic therapies [25–28], it is believed that knowledge of effective target biomarkers as well as the development of high-specificity epigenetic-modifying agents could lead to similar successes for non-cancerous conditions such as SS [22, 29, 30].

## Materials and methods

### Study subjects and clinical evaluation

A total of 131 female, non-Hispanic white individuals were selected from SICCA for this study. Multidimensional scaling (MDS) of genotype data confirms their non-Hispanic white ancestry and suggests that the majority of individuals are predominantly of French or Orcadian ancestry (S1 Fig). All individuals from the SICCA registry exhibited at least one symptom

Table 1. Summary statistics of SS phenotypes, potential confounders, and co-morbidities.

| | cases (n = 64) | non-cases (n = 67) | p-value |
|---|---|---|---|
| Focus score | 3.39 (1.83) | 0.89 (0.67) | 6.80E-6 |
| Left ocular staining score | 7.46 (2.91) | 3.19 (2.75) | 8.54E-12 |
| Right ocular staining score | 7.19 (3.17) | 3.25 (2.74) | 4.21E-10 |
| SSA seropositive (indicator) | 0.63 | 0 | 3.61E-14 |
| SSB seropositive (indicator) | 0.55 | 0 | 6.26E-12 |
| Unstimulated whole salivary flow rate | 0.34 (0.39) | 0.70 (0.54) | 8.20E-6 |
| Schirmer $\leq$ 5 mm/5min on at least one eye | 0.23 | 0.07 | 2.16E-2 |
| Self-reported age of SS onset at screening | 49.12 (10.80) | 46.10 (8.86) | 2.04E-1 |
| Censored age at study visit | 54.69 (11.94) | 53.46 (10.82) | 4.53E-1 |
| Current smoker | 0.01 | 0.06 | 3.90E-1 |
| Anticholinergic drug use | 0.40 | 0.51 | 2.76E-1 |
| SLE suspected | 0 | 0 | NA |
| SLE physician confirmed | 0.05 | 0.04 | 1.00 |
| RA suspected | 0 | 0.01 | 1.00 |
| RA physician confirmed | 0.06 | 0.03 | 6.34E-1 |

Means and corresponding standard deviations (in parenthesis) are reported for continuous variables, and proportions are reported for binary variables. The p-value reports significance of difference between cases and non-cases for a given variable, determined either with Wilcoxon's rank sum test for continuous variables or chi-square test of independence for binary variables. Missing values are excluded from summary statistics. NA = not available; SLE = systemic lupus erythematosus; RA = rheumatoid arthritis.

related to SS, specifically symptoms of dry eyes or dry mouth, prior suspicion/diagnosis of SS, positive serum anti-SSA, anti-SSB, rheumatoid factor or antinuclear antibody results, increase in dental caries, bilateral parotid gland enlargement, or a possible diagnosis of secondary SS [31]. Table 1 summarizes the SS phenotypes, potential confounders, and co-morbidities of these study subjects. Case status was determined according to the 2016 American College of Rheumatology/European League Against Rheumatism (ACR/EULAR) criteria for SS [32]. Non-cases from the SICCA registry with at least one, but not all, SS symptoms or signs were also included. More specifically, non-cases did not meet ACR/EULAR for SS but were enrolled in SICCA due to the presence of 1 or more symptoms or signs suggesting possible SS. Based on these criteria, we studied 64 SS cases and 67 non-cases.

## Methylotyping and data processing

DNA was extracted from the LSG tissue collected from each study subject as previously described [20]. DNA methylation was measured for each subject using the Illumina 450K Infinium Methylation BeadChip (450K) platform for 28 subjects and the Infinium MethylationEPIC (EPIC) platform for 103 subjects. The 450K and EPIC chips allow for high-throughput interrogation of more than 450,000 and 850,000 highly informative CpGs sites respectively, spanning ~22,000 genes across the genome.

Methylation data processing was performed using *Minfi*, a Bioconductor package for the analysis of Infinium DNA methylation microarrays [33]. Background subtraction with dye-bias normalization was performed on methylated and unmethylated signals with the *noob* procedure, followed by quantile normalization with *preprocessQuantile* [34, 35].

For joint analysis of all 131 samples, the intersection of CpGs from 450K and EPIC chips was selected for analysis, resulting in a starting number of 452,832 CpGs. Probes where more

than 5% of samples had a detection p-value > 0.01 were removed, to retain probes where signal is distinguishable from negative control probes. To remove probes with ambiguous methylation measurements due to incomplete binding between the DNA strand of interest and probe strand DNA, probes with SNPs with minor allele frequency greater than 0% at either the probe site, CpG interrogation site, or single nucleotide extension were removed. Finally, probes identified with probe-binding specificity and polymorphic targets problems, or cross-reactive probes, were removed [36, 37]. The final processed dataset consisted of 336,040 CpG sites. Since no subject had more than 5% of probes with detection p-value > 0.01, all 131 subjects were retained. Both M-values and β-values were used in subsequent analyses (see S1 Text).

## Removing unwanted DNA methylation variation

We identified array type (450K or EPIC), genetic ancestry, self-reported age of SS syndrome onset, collection phase, smoker status, anticholinergic drug use, and co-morbidities as potential confounders (Table 1). Of these, array type and genetic ancestry were found to be strongly associated with DNA methylation and case status respectively ($p \leq 0.05$) (S1 and S2 Figs), and analytical models were adjusted accordingly. However, case status was not associated with array type, because the distribution of cases and non-cases were similar between 450K and EPIC with 46.4% cases and 50.0% non-cases respectively (S2 Fig). Wilcoxon's rank sum test of difference in ancestry MDS component values between cases and non-cases revealed a significant association at p-value $\leq 0.05$ for components 2–4 and at p-value $\leq 0.10$ for component 1. Unwanted methylation variation due to array type and genetic ancestry (batch effects) were removed from β-values and M-values using *ComBat* from the *SVA* package, which applies an empirical Bayes, model-based location/scale batch adjustment [38, 39]. See S1 Text for details of *Combat* usage.

## Genotyping and quality control

The subject genotypes were taken from the genotypes of the larger SICCA cohort, which was genotyped on the Illumina HumanOmni2.5-4v1 or Illumina HumanOmni25M-8v1-1 arrays from DNA extracted from whole blood. All quality control steps performed have been previously described [7]. The final genotype dataset consisted of 1,392,448 SNPs.

## Dimensionality reduction

Principal component analysis (PCA) was performed on the centered and scaled β-value matrix $X \in \mathbb{R}^{n \times p}$, where $n$ and $p$ are the number of subjects and CpG sites, respectively. PCA was performed on methylation data prior and after batch correction with *ComBat*.

Multidimensional scaling (MDS) was performed to detect population structure using lower dimensions that explain observed genetic distance. With genotype data as reference allele counts, pairwise genotype dissimilarity is summarized by the distance matrix $D = J - IBS \in \mathbb{R}^{n \times n}$, where $IBS \in \mathbb{R}^{n \times n}$ is the identity-by-state similarity matrix and $J \in \mathbb{R}^{n \times n}$ is the all-ones matrix. MDS of genotypes from the 131 subjects and reference European sub-populations from the Human Genome Diversity Project (HGDP) [40] was performed using PLINK 1.9 to assess association between genetic ancestry and case-control status [41].

## Identification of differentially methylated regions

Differentially-methylated regions (DMRs) were identified using *bumphunter*, which searches for bumps, or contiguous CpG sites consistently hypermethylated or hypomethylated in one

group of subjects compared to the other [42]. The linear regression specified for *bumphunter* was

$$M \sim outcome + array\ type + C1 + \cdots + C5, \tag{1}$$

which controlled for array type and genetic ancestry. Here, "M" is the M-value without batch correction with *Combat*, *outcome* is SS case status, *array type* indicates array (450K or EPIC), and $C1 - C5$ indicate the first five MDS components of genotype data. The number of bootstrap resampling *B* was set to 1,000 for generating null distribution of candidate DMRs for establishing significance. Significant SS DMRs were stringently selected as those with, *fwerArea* ≤ 0.05 defined as proportion of bootstraps with maximum bump area greater than observed DMR area, and consists of at least two CpG sites. See S1 Text for details on choice of *bumphunter* hyperparameters and annotation of DMRs.

## Gene set enrichment analysis

Since methylation at transcription start sites and gene bodies has been shown to regulate gene expression [43], we restricted gene set enrichment analysis (GSEA) to genes differentially methylated at the promoter or gene body. DMR genes were tested for enrichment of gene ontology (GO) gene sets from the Molecular Signatures Database [44] combined with SS-related gene sets from past studies using the hypergeometric test (see S1 Text for gene set details). False discovery rate was controlled with the Benjamini-Hochberg procedure [45]. Since genes in the same pathway tend to be up or down-regulated together, GSEA was performed separately for hypermethylated and hypomethylated DMR genes in cases compared to non-cases [46].

## Identification of DNA methylation quantitative trait loci

Methylation quantitative trait loci (meQTLs) are loci whose genotypes are associated with DNA methylation. We test for short-range *cis*-meQTLs, defined as SNPs in the ±250 kb genomic region from the DMR start and end positions. This window size was chosen based on previous meQTL studies of similar sample sizes to roughly ensure adequate power [19, 47–50]. Although long-range meQTL effects spanning several megabase pairs (mb) has been observed at the MHC [23], McRae *et al.* observed most significant meQTLs are within 100 kb of target CpGs in their study involving a window size of ±2 mb [50]. Thus, we do not expect many such meQTLs to be missed if they exist. SNPs in approximate linkage equilibrium were selected using PLINK as those satisfying pairwise correlation $R^2 \leq 0.5$ in a 250,000 bp window, with a window stride of 25,000 bp [41]. The association between a candidate meQTL and DMR was established by regressing the M-value, averaged across CpG sites of the DMR, against genotype encoded as 0, 1, or 2 copies of the reference allele, from all 131 subjects. The DNA methylation values used for identifying meQTLs were batch-corrected for array type and genetic ancestry. Significance of association was evaluated using *t*-test from linear regression. False discovery rate was controlled with the Benjamini–Hochberg procedure [45].

## Mediation analysis with causal inference test

We used the causal inference test (CIT) to determine whether DNA methylation mediates genetic risk by evaluating statistical evidence for a causal mediation model [24, 51]. Specifically, the CIT evaluates a set of statistical tests of the necessary and sufficient conditions for the causal mediation relationship involving genotype "G", DNA methylation "M", and case status "S". In the causal graph of this causal mediation model, the directed edge travels from "G" to "S" through "M". The conditions are:

1. $S \sim G$

2. $S \sim M \mid S$

3. $M \sim S \mid G$

4. $S \perp G \mid M,$

where "$\sim$" denotes associated with and "$\perp$" denotes independent of. In the event of reverse causation, where the disease condition induces differential methylation, a spurious association will instead be observed between genotype and SS, failing condition four. The maximum p-value from these four statistical tests is the CIT p-value. See Millstein *et al.* for additional details on the CIT [24]. The CIT was performed for the identified meQTL-DMR pairs using genotype, DNA methylation, and SS case status from all 131 subjects. The genotype and DNA methylation data are encoded the same way as for the identification of meQTLs. The CIT genotype is encoded as 0, 1, or 2 copies of the reference allele, DNA methylation value is the batch-adjusted M-value, and SS is binary case status. False discovery rate was controlled at or under 5% using the permutation-based q-value developed and implemented by Millstein *et al.* [51, 52]. See S1 Text for usage details of the CIT.

## Ethics statement

This study was approved by the Institutional Review Board of the Human Research Protection Program at the University of California, San Francisco (approval number: 10–02551).

## Results

### Characterization of SS cases and non-cases

We start by characterizing the clinical and global DNA methylation profiles of SS cases and non-cases. Although all non-cases exhibit at least one SS-related phenotype, cases have significantly higher focus scores, ocular staining scores, SSA and SSB seropositivity, Schirmer test positivity rate, and lower unstimulated whole salivary flow rates (Table 1). This is expected, since severity in these phenotypes is the basis upon which the 2016 ACR/EULAR criteria classifies SS [32]. From Table 1, there are no significant differences in the potential confounders of age-related variables, smoking habits, and anticholinergic drug use. Around 5% of cases and non-cases have physician confirmed co-morbidities of systemic lupus erythematosus or rheumatoid arthritis, without significant differences in occurrence between the groups. Thus, the presence of co-morbidities is unlikely to significantly influence our differential methylation analysis results. PCA of adjusted DNA methylation data shows clear global differences between cases and non-cases (S3 Fig). This difference is immediately seen in the first principal component, which explains the most variance of the projected methylation data. This highlights the relevance of DNA methylation differences in the context of SS and LSG.

### Hypomethylation of genes involved in immune response

Analysis with *Bumphunter* identified 215 significant DMRs from 2,747 candidate "bumps" (S1 Table). Of the 215 DMRs, 169 were hypermethylated regions and 46 were hypomethylated regions, in cases relative to non-cases. Approximately 84% of DMRs were located in either promoters or gene bodies (Fig 1A), locations where differential methylation tends to influence transcription [43]. The top three DMR-contributing chromosomes were chromosomes 1, 6, and 17, and a majority of DMRs on chromosome 6 overlapped or surrounded the MHC (Fig 1B). Detailed annotation of significant DMRs are in S1 Table. We found no overlap between

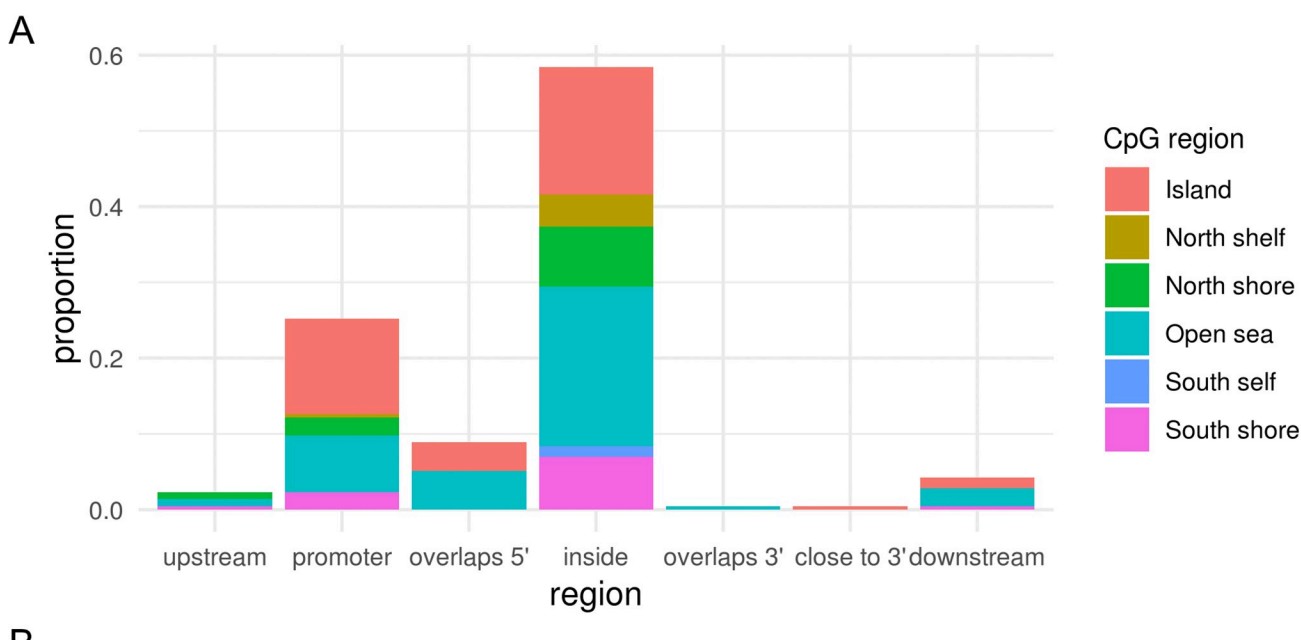

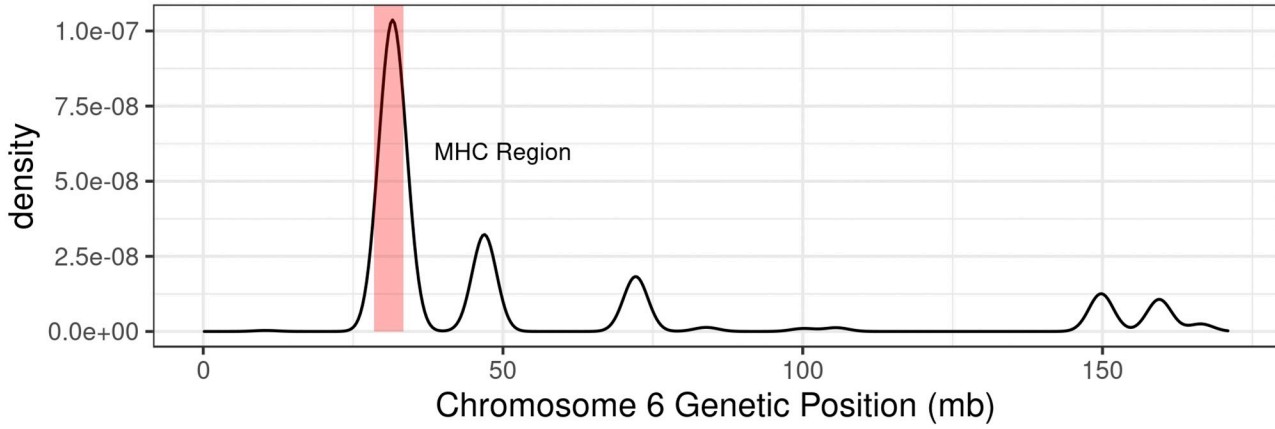

**Fig 1. DMR characteristics.** (A) Proportion of SS DMR locations relative to closest gene, and CpG type proportions at each DMR location; most DMRs are located either in the gene body (inside) or promoter, and most DMR CpG sites are either in the CpG island or the open sea. (B) Density plot of SS DMR locations on chromosome 6, where a DMR's location is represented by GRCh37 genetic coordinates of its first CpG site to last CpG site. The shaded red region denotes the MHC region (28,477,797 bp—33,448,354 bp on chromosome 6). mb = megabase pairs.

these DMRs and gene regions with established or suggestive association with SS, even at the MHC [7, 8, 53]. We define an overlap to occur when the genetic coordinate range (start to end) of a gene overlaps with that of the DMR. S2 Table lists the set of genes with which we examined overlap with DMRs. Although Miceli-Richard *et al.* observed an overlap between genetic risk loci for SS with differentially-methylated DNA regions, our studies differ in the target tissue involved and definition of differential-methylated regions (i.e. region vs single CpG site) [18].

Genes near hypomethylated regions in cases were enriched for gene sets associated with immune function (Table 2), with the top gene sets almost exclusively related to immune response. This was expected given many DMRs were concentrated at the MHC. *IRF5*, which resides on chromosome 7 and is the strongest genetic risk factor for SS outside the MHC [8],

**Table 2. Top gene sets enriched for hypomethylated genes in SS.**

| gene set | n | overlap genes | p-value | adj. p-value |
|---|---|---|---|---|
| SS DMP genes | 8 | *TAP1, LTA, PSMB8, AIM2, NCKAP1L, LINC00426, LCP2, ARHGAP25* | 3.80E-18 | 1.71E-14 |
| Antigen processing and presentation of endogenous peptide antigen | 4 | *HLA-E, HLA-B, TAP1, ABCB1* | 1.60E-12 | 3.59E-9 |
| Antigen processing and presentation of peptide antigen via MHC class I | 6 | *PSMB9, HLA-E, PSMB8, HLA-B, TAP1, ABCB1* | 4.74E-12 | 5.53E-9 |
| Antigen processing and presentation of endogenous antigen | 4 | *HLA-E, HLA-B, TAP1, ABCB1* | 4.92E-12 | 5.53E-9 |
| Negative regulation of innate immune response | 4 | *HLA-E, HLA-B, TAP1, NLRC5* | 3.93E-10 | 3.24E-7 |
| Negative regulation of natural killer cell mediated immunity | 3 | *HLA-E, HLA-B, TAP1* | 4.32E-10 | 3.24E-7 |
| Antigen processing and presentation via MHC class IB | 3 | *HLA-E, TAP1, ABCB1* | 1.19E-10 | 7.64E-7 |
| Positive regulation of antigen processing and presentation | 3 | *ABCB1, CCR7, TAP1* | 1.58E-9 | 7.92E-7 |
| Positive regulation of humoral immune response | 3 | *LTA, TNF, CCR7* | 1.58E-9 | 7.92E-7 |
| Negative regulation of cell killing | 3 | *HLA-B, HLA-E, TAP1* | 2.66E-9 | 1.20E-6 |

Candidate gene sets include GO gene sets from the Molecular Signatures Database [44], a set of genes previously reported to harbor differentially methylated CpG sites between SS cases and non-cases (SS DMP genes) [20], and a set of genes previously reported to be differentially expressed between SS cases and healthy controls (SS DE genes) [54]. n = number of overlapping genes; adj. p-value = Benjamini-Hochberg adjusted p-value.

was not the nearest gene for any DMRs. Of the 131 individuals in our study, 26 were in a previous LSG study by Cole *et al.*, which identified 57 genes whose promoters were hypomethylated in SS relative to controls [20]. From GSEA, these 57 genes (SS DMP genes) form the top enriched gene set with an adjusted p-value of 1.71E-4 (Table 2). Finally, the DMR gene *PSMB9* was one of the 45 genes that previously demonstrated differential expression between SS cases and non-cases [54].

In contrast to hypomethylated regions, genes near hypermethylated regions were enriched for gene sets with several functions; therefore, the overall picture for hypermethylation in cases was less clear. Table 3 shows that the top gene sets were associated with nervous system development and cellular transport and signaling.

## DNA methylation mediates the effect of meQTLs on SS at the MHC

We tested for association between average DMR methylation M-values and SNPs in approximate linkage equilibrium in a ±250kb neighborhood of each DMR, which yielded 20,754 unique DMR-SNP candidate pairs to test. A total of 26 meQTL-DMR associations were identified under the Benjamini-Hochberg adjusted p-value cutoff of 0.05, with one each from chromosomes 3, 11, 12, 16, and two from chromosome 4; the rest were located within the MHC region on chromosome 6 (S3 Table). Fig 2A shows how methylation levels vary by genotype for example meQTL rs9275224. Note that a meQTL can be associated with multiple DMRs, and a DMR can be associated with multiple meQTLs. Down-sampling SNPs at the MHC to achieve comparable SNP densities to that of non-MHC regions still resulted in a higher meQTL discovery rate at the MHC relative to non-MHC regions (see Supplementary Results in S1 Text for more details). Thus, the higher discovery rate at the MHC cannot be explained by higher SNP densities. The distribution of meQTL-DMR distances is concentrated around 160 kb, with an average of 153 kb, which is well within the limit of 250 kb (Fig 2B). Thus, the

**Table 3. Top gene sets enriched for hypermethylated genes in SS.**

| gene set | n | overlap genes | p-value | adj. p-value |
|---|---|---|---|---|
| Positive regulation of transporter activity | 6 | *WNK4, ATP1B2, RELN, HAP1, CACNB2, TRPC6* | 1.36E-8 | 6.12E-5 |
| Diencephalon development | 5 | *ETS1, GSX1, GLI2, HAP1, SLC6A4* | 4.17E-7 | 9.38E-4 |
| Hypothalamus development | 3 | *ETS1, GSX1, HAP1* | 1.73E-6 | 2.59E-3 |
| Vasoconstriction | 3 | *EDN3, HTR1A, SLC6A4* | 3.29E-6 | 3.42E-3 |
| Modulation of excitatory postsynaptic potential | 3 | *ZMYND8, CELF4, RELN* | 4.38E-6 | 3.42E-3 |
| Somatic stem cell population maintenance | 4 | *WNT98, LRP5, PBX1, BCL9* | 4.59E-6 | 3.42E-3 |
| Nerve development | 4 | *HOXB3, COL25A1, TFAP2A, SLITRK6* | 5.32E-6 | 3.42E-3 |
| Peptide Transport | 4 | *EDN3, SLC15A2, FAM3B, TAPBP* | 7.06E-6 | 3.97E-3 |
| Anatomical structure regression | 2 | *LRP5, GLI2* | 1.03E-5 | 4.86E-3 |
| ERBB2 signaling pathway | 3 | *ERBB2, GRB7, SHC1* | 1.28E-5 | 4.86E-3 |

Candidate gene sets include GO gene sets from the Molecular Signatures Database [44], a set of genes previously reported to harbor differentially methylated CpG sites between SS cases and non-cases (SS DMP genes) [20], and a set of genes previously reported to be differentially expressed between SS cases and healthy controls (SS DE genes) [54]. n = number of overlapping genes; adj. p-value = Benjamini-Hochberg adjusted p-value.

window size of ±250kb appears sufficient for identifying most *cis*-meQTLs. While the density plot of the meQTL-DMR distances appears somewhat bimodal, the smaller peak at around 60 kb is most likely an artifact due to small sample size and the smoothing process of a density plot. From S3 Table, there are only 3 meQTL-DMR distances ranging from 75 kb to 80 kb.

Of these 26 meQTL-DMR pairs, the CIT identified 19 with significant evidence supporting the causal mediation model (q-value ≤ 0.05); one pair each was from chromosomes 3, 12, and 16, and the rest were from chromosome 6 (Table 4). At the MHC, the region spanning the *HLA-DQA1*, *HLA-DQB1*, and *HLA-DQA2* loci contained a high density of DMR-meQTL pairs, with five DMRs and four meQTLs (Fig 2C). In total, the meQTL-DMR pairs from Table 4 represent 12 unique DMRs and 9 unique SNPs. The remainder of the 26 associated meQTL-DMR pairs did not support the causal mediation model, with the three unique DMRs potentially consequences of reverse causation (S3 Table). The remaining 200 of the 215 DMRs discovered were not associated with any nearby SNPs (S3 Table); thus, no evidence of nearby genetic control was detected, and it is still unknown which ones represent potential cases of reverse causation.

Utilizing data from a previous genome-wide association study (GWAS) of SS involving 2,131 European individuals [7], we tested the association with SS for all meQTLs supporting the causal mediation model (Table 4), using the updated 2016 ACR/EULAR classification criteria to define cases and controls [32]. European ancestry, sex, and smoking status were adjusted as described in Taylor *et al.* [7]. Table 5 shows these association results. Of these, five meQTLs at the MHC from 31,603 kb to 32,681 kb reached genome-wide significance (Fig 3 and Table 5). MeQTLs supporting the causal mediation model from chromosomes 3, 12, and 16 are not significantly associated with SS, with p-values not even satisfying the significance level of a single hypothesis test (p-value ≤ 0.05).

We next examined the extent to which linkage disequilibrium (LD) can explain the association of meQTLs with SS at the MHC. We obtained squared coefficient of correlation statistics ($R^2$) as a measure of LD based on genotypes of European populations from the 1000 Genomes Project (S4 Table) [55]. Fig 4A shows the LD heatmap among the six meQTLs at the MHC, and Fig 4B shows the LD heatmap between the six meQTLs and MHC SNPs that previously demonstrated association with SS in Europeans [7, 8]. These meQTLs are in mild LD with each other (Fig 4A), with a maximum $R^2$ of 0.357 (S4 Table). This is expected, since we pre-

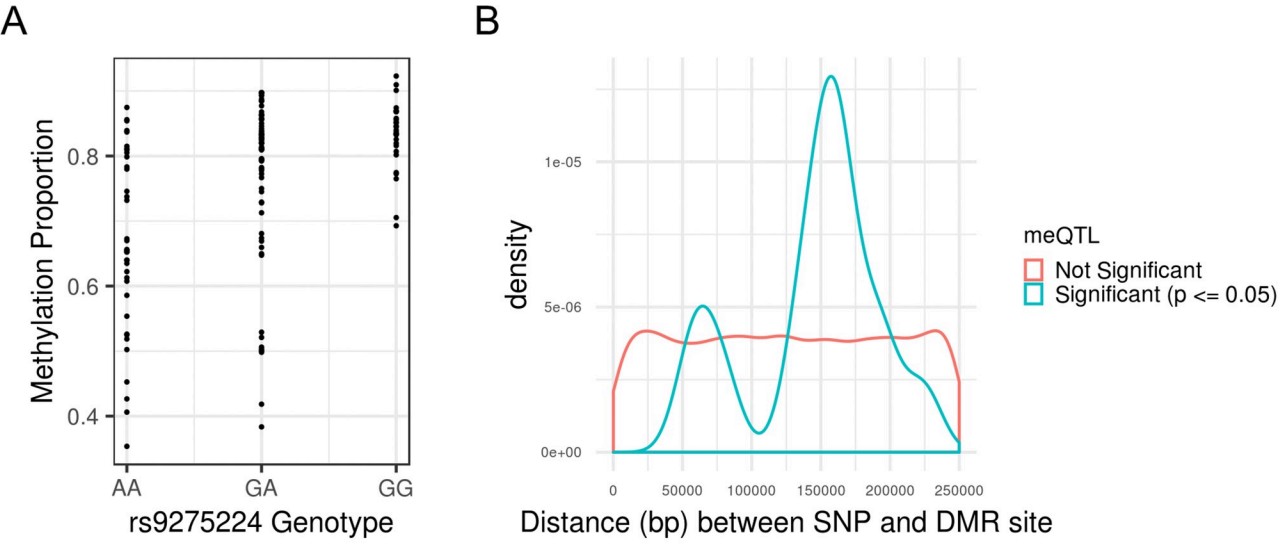

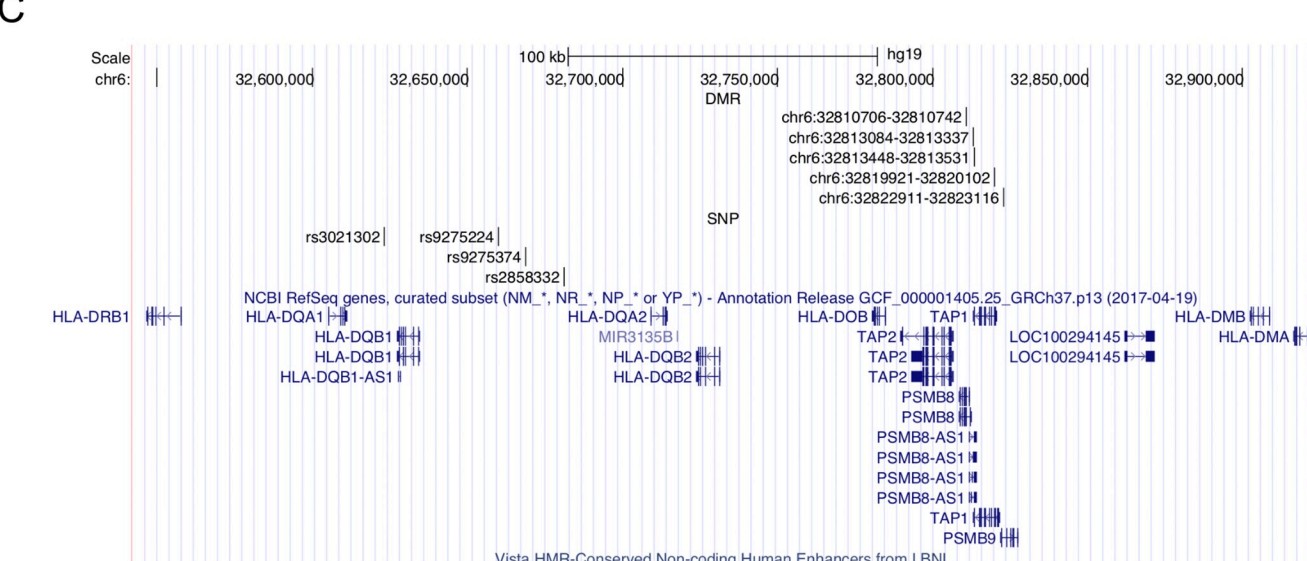

**Fig 2. MeQTLs associated with SS DMR methylation M-values.** (A) SNP rs9275224 is a meQTL associated with average M-value of the DMR at genetic positions 32,810,706–32,810,742 (GRCh37) on chromosome 6. See S3 Table for the remaining meQTL-DMR pairs. (B) Density plot of associated and unassociated SNP-DMR pairs by absolute distance. The significance criteria for association is having a Benjamini-Hochberg adjusted p-value (p) ≤ 0.05. While distance is approximately uniformly distributed for unassociated SNP-DMR pairs, the distances of associated SNP-DMR pairs is concentrated around 153 kb. (C) MHC region spanning the *HLA-DQA1*, *HLA-DQB1*, and *HLA-DQA2* loci with high density of the meQTL-DMR pairs. Each DMR is specified by its chromosome, starting position, and ending position, in GRCh37 genetic coordinates.

selected SNPs in approximate linkage equilibrium before searching for meQTLs. Using multi-variate logistic regression modeling and adjusting for European ancestry, sex, and smoking status as described in Taylor *et al.* [7], we found modest evidence that the meQTLs rs3021302, rs9275224, and rs2858332 exhibit independent effects (p-value ≤ 0.05; Table 6).

However, Fig 4B shows that these meQTLs exhibiting independent effects are in stronger LD with some SS SNPs. Considering $R^2 > 0.50$ as reflecting at least modest LD, the meQTL rs2858332 is in relatively strong LD with rs9275572 ($R^2 = 0.741$), which is in the gene regions *HLA-DQB1* and *HLA-DQA2* [7]. Although meQTL rs9275224 is not in as strong LD with

**Table 4. Top causal inference test results for meQTLs of SS DMRs.**

| SNP rs ID | SNP position | A1 | A2 | SS DMR | distance | p.cit | q.cit |
|---|---|---|---|---|---|---|---|
| rs9275224 | 32659878 | G | A | chr6:32810706–32810742 | 150828 | 1.00E-3 | 2.11E-3 |
| rs9275224 | 32659878 | G | A | chr6:32819921–32820102 | 160043 | 1.00E-3 | 2.11E-3 |
| rs9275224 | 32659878 | G | A | chr6:32822911–32823116 | 163033 | 1.00E-3 | 2.11E-3 |
| rs9275224 | 32659878 | G | A | chr6:32813084–32813337 | 153206 | 1.00E-3 | 2.11E-3 |
| rs9275224 | 32659878 | G | A | chr6:32813448–32813531 | 153570 | 1.00E-3 | 2.11E-3 |
| rs2261033 | 31603591 | G | A | chr6:31544694–31544931 | 58660 | 1.17E-3 | 2.11E-3 |
| rs2261033 | 31603591 | G | A | chr6:31527920–31528239 | 75352 | 1.89E-3 | 2.11E-3 |
| rs2734985 | 29818662 | G | A | chr6:30042980–30042985 | 224318 | 1.99E-3 | 2.11E-3 |
| rs9275374 | 32668526 | A | G | chr6:32810706–32810742 | 142180 | 3.99E-3 | 3.47E-3 |
| rs2261033 | 31603591 | G | A | chr6:31539973–31539998 | 63593 | 5.25E-3 | 4.17E-3 |
| rs13335209 | 87860446 | A | C | chr16:87636539–87636594 | 223852 | 5.78E-3 | 4.30E-3 |
| rs3021302 | 32623150 | G | A | chr6:32810706–32810742 | 187556 | 7.84E-3 | 4.89E-3 |
| rs3021302 | 32623150 | G | A | chr6:32819921–32820102 | 196771 | 1.47E-2 | 9.29E-3 |
| rs2858332 | 32681161 | C | A | chr6:32819921–32820102 | 138760 | 1.63E-2 | 1.05E-2 |
| rs17407659 | 24238010 | A | G | chr12:24104007–24104115 | 133895 | 1.74E-2 | 1.35E-2 |
| rs3021302 | 32623150 | G | A | chr6:32813084–32813337 | 189934 | 2.49E-2 | 1.64E-2 |
| rs3021302 | 32623150 | G | A | chr6:32822911–32823116 | 199761 | 2.69E-2 | 1.74E-2 |
| rs2858332 | 32681161 | C | A | chr6:32810706–32810742 | 129545 | 3.36E-2 | 2.14E-2 |
| rs76027985 | 112439220 | G | A | chr3:112359488–112359557 | 79663 | 3.65E-2 | 2.44E-2 |

All genetic positions are based on GRCh37 coordinates, and DMRs are denoted by the chromosome, start position, and end position. Distance refers to base pair distance between DMR and meQTL. A1 = allele 1; A2 = allele 2; SS DMR = differentially-methylated regions for Sjögren's syndrome; p.cit = causal inference test p-value; q.cit = permutation-based q-values from the causal inference test.

rs9275572 as meQTL rs2858332 ($R^2$ = 0.446), rs9275572 is still the SNP that rs9275224 is in strongest LD with. Lastly, meQTL rs3021302 is in modest LD with rs115575857 and rs3129716 (both $R^2$ = 0.572), which are in the gene region *HLA-DQB1* [8]. Based on the LD statistics, these meQTLs likely do not tag the SS SNPs we compared with, but may reflect association of different HLA alleles with SS given modest evidence of independent effects.

**Table 5. Association of meQTLs with SS in European GWAS.**

| SNP rs ID | CHR | genetic position | p-value | OR (95% CI) |
|---|---|---|---|---|
| rs76027985 | 3 | 112439220 | 9.02E-1 | 1.041 (0.549–1.973) |
| rs2734985 | 6 | 29818662 | 1.17E-3 | 1.274 (1.101–1.474) |
| rs2261033 | 6 | 31603591 | 2.62E-12 | 0.617 (0.539–0.707) |
| rs3021302 | 6 | 32623150 | 2.21E-29 | 2.475 (2.114–2.898) |
| rs9275224 | 6 | 32659878 | 5.01E-21 | 1.937 (1.688–2.224) |
| rs9275374 | 6 | 32668526 | 1.62E-9 | 0.598 (0.506–0.707) |
| rs2858332 | 6 | 32681161 | 9.02E-25 | 2.071 (1.803–2.380) |
| rs17407659 | 12 | 24238010 | 2.63E-1 | 0.889 (0.723–1.092) |
| rs13335209 | 16 | 87860446 | 5.63E-1 | 1.039 (0.912–1.185) |

Association results of meQTLs that support causal mediation model in previous European GWAS for SS [7]. SS case status was determined based on the 2016 ACR/EULAR classification criteria [32]. The genome-wide significance threshold is p-value $< 5 \times 10^{-8}$. CI = confidence interval; CHR = chromosome.

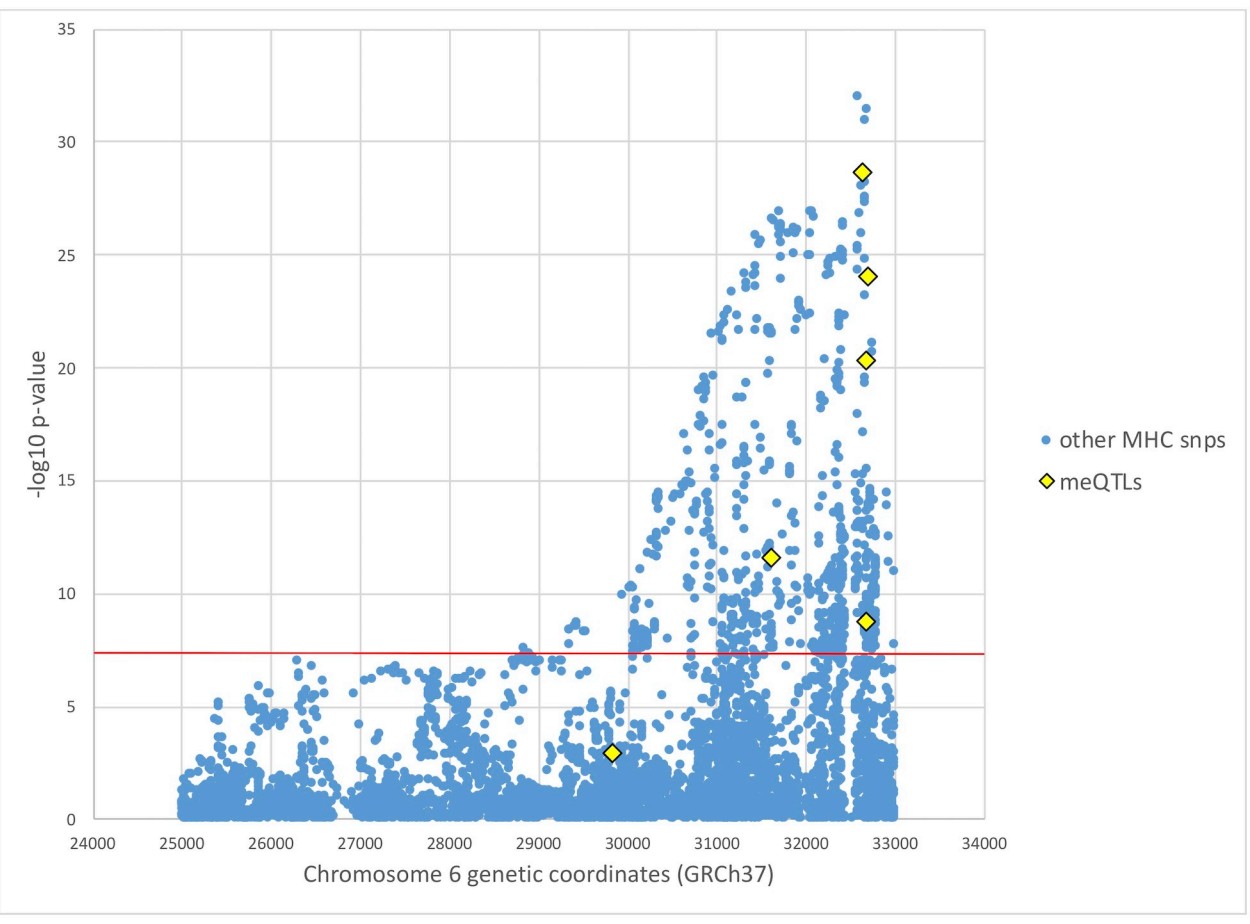

**Fig 3. Manhattan plot of SS GWAS results at the MHC.** European SS GWAS results at the MHC from Taylor *et al.* [7], with mediating meQTL p-values from this study colored in yellow. SS case status was determined based on the 2016 ACR/EULAR classification criteria [32]. The red horizontal line indicates genome-wide significance level of p-value $< 5 \times 10^{-8}$.

## Discussion

We investigated the relationship between genetic variation, DNA methylation, and SS in the largest study of LSG, to date. We compared SS cases against symptomatic non-cases, and results show that significant differential methylation in LSG exists and is primarily driven by case status. Results from DMR analysis of LSG are consistent with the general theme of hypo-methylation previously reported in a much smaller sample [20], providing strong support for these findings. We applied the CIT to genotype and DNA methylation data from the same individuals, and conclude that genetic control of differential methylation is a risk factor for SS, especially at the MHC.

General hypomethylation of genomic regions involved in the immune response in LSG remains one of the most significant findings (Table 2), with many DMRs located in the MHC region. Many of these hypomethylated genes have biological roles closely related to SS patho-physiology. For example, dendritic cells in the glands produce high levels of interferons [1], and *PSMB8* and *PSMB9*, whose expressions are induced by gamma interferon, were both hypomethylated in SS cases compared to non-cases. Genes *PSMB8* and *PSMB9* encode cata-lytic subunits of the immunoproteasome that is involved in peptide presentation on the surface of antigen-presenting cells [56]. Hypomethylation of *PSMB9* may have a causal role in

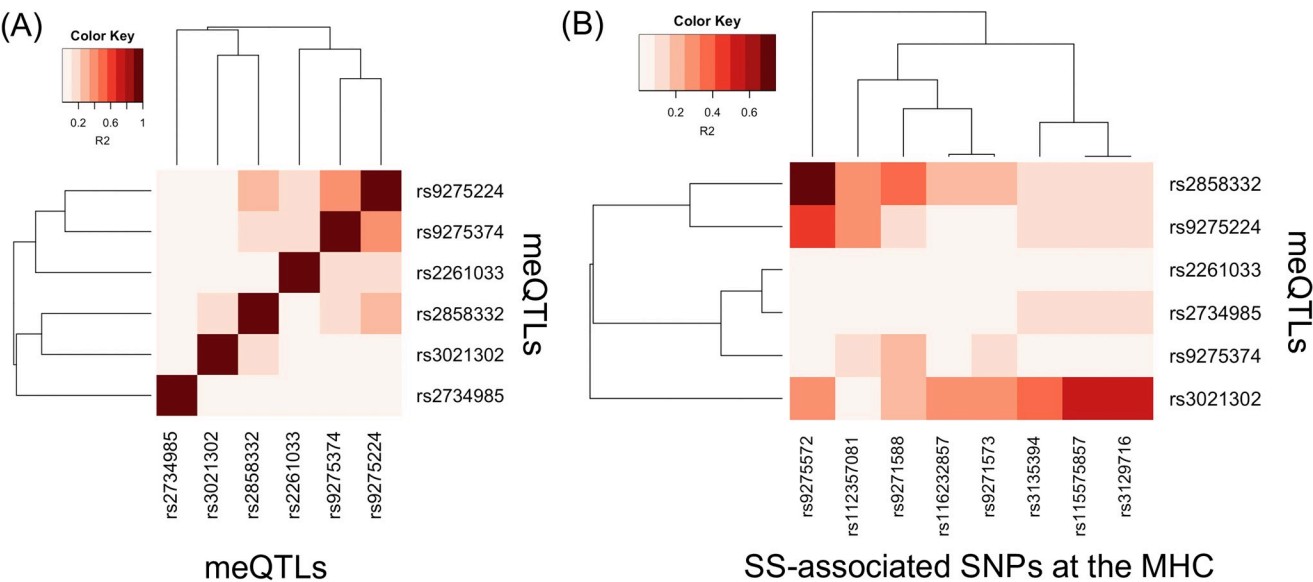

**Fig 4. LD heatmap for MHC meQTLs supporting the causal mediation model.** Heatmap of the LD measure of $R^2$ statistics, based on European populations from the 1000 Genomes Project [55]. (A) LD among meQTLs and (B) LD between meQTLs and established SS-associated SNPs at the MHC, for Europeans [7, 8].

increasing expression levels in SS [54]. Previous studies have suggested that differential DNA methylation in SS could be controlled by B cells infiltrating the LSG, which in turn may affect the expression of inflammatory genes [13, 14].

Although the overall picture for hypermethylated regions in cases is less clear than that for hypomethylated regions, gene set enrichment analysis (GSEA) suggested some degree of neurological involvement in SS (Table 3). Peripheral neuropathy is the most common neurological complication in SS, but involvement of the central nervous system has also been observed, including cognitive disorder meningitis and optic neuritis [57]. The pathological mechanism by which SS leads to damage of the nervous system is not well-established, but it is thought to involve inflammatory infiltration of the dorsal root ganglia [1, 57]. Although dryness of mouth resulting from reduced saliva flow is negatively correlated with glandular innervation in general [58], another study found no differences in innervation pattern between SS cases and healthy controls, and found both groups to have functional acinar receptor systems [59].

**Table 6. Multivariate logistic regression of SS status against MHC meQTLs.**

| SNP rs ID | CHR | genetic position | p-value | OR (95% CI) |
|---|---|---|---|---|
| rs2734985 | 6 | 29818662 | 0.505 | 0.952 (0.823–1.101) |
| rs2261033 | 6 | 31603591 | 0.072 | 0.875 (0.756–1.010) |
| rs3021302 | 6 | 32623150 | 0.000 | 1.688 (1.413–2.015) |
| rs9275224 | 6 | 32659878 | 0.014 | 1.257 (1.048–1.509) |
| rs9275374 | 6 | 32668526 | 0.519 | 1.066 (0.878–1.294) |
| rs2858332 | 6 | 32681161 | 0.002 | 1.304 (1.102–1.544) |

Multivariate logistic regression of SS case status again all MHC MeQTLs supporting the causal mediation model based on genotypes from previous European GWAS [7]. The logistic regression adjusted for European ancestry, sex, and smoking status, following Taylor *et al.* [7], and SS case status was determined based on the 2016 ACR/EULAR classification criteria [32]. CHR = chromosome; OR = odds ratio; CI = confidence interval.

Evidence of allele-specific methylation over extended genomic regions has been previously reported and can vary by tissue, developmental stage, and ancestry [47]. Here, we identified DMRs in SS whose methylation levels appear to be under genetic control using the CIT. Twelve of the 215 DMRs demonstrated evidence of causal dependence on neighboring genotypes, with the majority residing in the MHC. Furthermore, 9 of the 16 DMRs in the MHC region showed evidence of mediation, supporting a general theme of genetic control of DNA methylation at the MHC. Majority of these MHC meQTLs involved in this causal mediation relationship are significantly associated with SS based on a previous GWAS for Europeans [7]. Our analysis shows modest evidence that some of these meQTLs have independent effects on SS risk, and that these meQTLs are in modest LD with some, but not all, established risk alleles in the HLA gene regions [7, 8]. However, larger studies are likely needed to determine whether the association of HLA alleles with SS is also mediated by DNA methylation, due to the polymorphic nature of HLA alleles. Using a combined genetic and epigenetics approach, our results support a role for functional relevance of previously established SS-associated SNPs at the MHC.

Findings that DNA methylation can mediate genetic risk conferred by the MHC, has been identified in a number of other autoimmune diseases. Differential methylation encompassing exon 2 of *HLA-DRB1*15:01* has been shown in monocytes to the mediate effect of the HLA-*DRB1*15:01* allele on its expression and risk of multiple sclerosis [60]. In psoriasis, the majority of reported meQTLs also reside in the MHC, although target CpG loci were located more than 500 kb away from their corresponding meQTLs. Using the CIT, 11 SNP-CpG pairs were found to exhibit a methylation-mediated relationship with psoriasis in skin tissue [61]. In rheumatoid arthritis, DNA methylation levels were found to mediate genetic risk within the MHC in whole blood [62]. Our results add to the growing evidence that the MHC likely confers genetic risk of disease in a more complex way than previously understood.

DNA methylation is currently thought to be influenced by genetic factors, age, environment and lifestyle, and tissue-type [63–66]. By identifying CpG sites that mediate nearby genetic risk for SS, CpG sites whose methylation levels may be altered by disease status, and CpG sites showing no evidence of nearby genetic control, we provide information that could be relevant for the potential therapeutic application of site-specific epigenetic editing for SS [67]. For example, it may be important to avoid targeting CpG sites whose methylation levels are altered by disease status. Currently, epigenetic therapy has been most effective for hematological malignancies but not in solid tumors [26, 28]. Epigenetic therapeutic approaches for other disease conditions remain in development, facing challenges such as lack of knowledge of effective target biomarkers, insufficient drug specificity, and dose-limiting toxicities [22, 28–30]. Nevertheless, autoimmune diseases have been cited as a promising area for the application of epigenetic therapies [22].

Since the LSG consists of a mixture of epithelial and inflammatory cells, a limitation of our study of LSG tissue is that it is unclear to what extent the observed methylation differences are explained by differences in cellular composition [30]. Without a reference dataset of methylation measurements on separated cell types from LSG, it is difficult to adjust for cell type heterogeneity using reference-based methods, which has been shown to perform better than reference-free methods [68]. Reference-free correction methods have been shown to vary widely in performance and lead to false positives in epigenome-wide association studies. A similar study of LSG has observed differentially-methylated cell differentiation markers as evidence for an increased proportion of immune cells [20], although we did not replicate these findings in our DMR analysis. Evidence of cell-specific differential methylation has been observed for salivary gland epithelial cells in SS [21]. Further investigation is needed to

establish the relative contributions of cell-specific differential methylation and cellular heterogeneity to differential methylation in LSG tissue.

In conclusion, we report evidence of genetic control of differential DNA methylation in SS by performing a formal CIT on genotype and DNA methylation datasets obtained from 131 individuals with LSG tissue and genotype data. We extended and replicated previous hypomethylation findings observed in many immune-related genes in SS cases, particularly those at the MHC. Our results also support the potential involvement of neurological processes in SS. By performing CIT on DMRs and their nearby meQTLs, we found that many DMRs associated with nearby risk alleles at the MHC were also mediators of SS risk. Interestingly, we did not observe as strong an evidence of mediation for SS DMRs at non-MHC locations. Through a formal study of the causal mediation relationship between genetic variation, DNA methylation, and SS case status, our findings provide essential information for the development of site-specific methylation-modifying therapies for SS.

## Supporting information

**S1 Fig. MDS of genotype data from SICCA study subjects and HGDP reference European samples.** Component 1 (C1) and component 2 (C2) refer to the two dimensions projected to by MDS.
(TIFF)

**S2 Fig. PCA of processed β-values, prior to batch-correction with *ComBat*.** The array type (450K or EPIC) for methylotyping is indicated by color. The array types 450K and EPIC show strong separation on PC2.
(TIFF)

**S3 Fig. PCA of processed β-values, after batch-correction with *ComBat*.** SS case status, as determined by the 2016 ACR/EULAR diagnostic criteria, is indicated by color [32]. Cases and non-cases show strong separation on PC1.
(TIFF)

**S4 Fig. Prior plot of kernel estimate of batch effect (black) and parametric estimate of batch effect (red) from *ComBat*.** (A) $\beta$-values and (B) M-values.
(TIFF)

**S5 Fig. Number of bumps found for SS and their sizes at different *bumphunter* coefficient cutoffs.** (A) Violin plot of bump sizes at each cutoff (B) Number of bumps discovered at each cutoff.
(TIFF)

**S1 Table. DMRs for SS and their annotations.** (A) The DMRs listed satisfy fwerArea $\leq$ 0.05 with least two CpG sites. The location relative to the DMR's closest gene are listed in the "gene" and "region" columns respectively. The column "value" is the average linear regression coefficients across DMR CpG sites, "area" is the absolute sum of linear regression coefficients for DMR CpG sites, "fwerArea" is the proportion of bootstraps with at least one candidate DMR area greater than observed DMR area, and "p.valueArea" is proportion of bootstraps with maximum bump area exceeding the observed area. For CpG site "island location", "N_Shore" = north shore, "S_Shore" = south shore, "N_Shelf" = north shelf, and "S_Shelf" = south shelf, and "OpenSea" = open sea. (B) CpG probes corresponding to each DMR.
(XLSX)

**S2 Table. Gene regions with established or suggestive associations with SS.**
(DOCX)

**S3 Table. DMR and associated meQTLs.** Statistical testing of association between average SS DMR methylation M-values and SNPs within 250 kb base pairs from the first and last CpG site of the DMR. DNA methylation values are batch-adjusted prior to testing and significance is established via the t-test from linear regression of M-values on copies of the reference allele. chr = chromosome; position = GRCh37 genetic coordinate of SNP; A1 = allele 1; A2 = allele 2; dmr = differentially-methylated region, represented by chromosome, start position, and end position; distance = base pair distance between SNP and DMR; coefficient = coefficient of allele copy number from linear regression; p_bh = Benjamini-Hochberg adjusted p-value.
(XLSX)

**S4 Table. Linkage disequilibrium statistics ($R^2$) regarding MHC meQTLs supporting the causal mediation model.** $R^2$ statistics are based on European populations from the 1000 Genomes Project [55]. (A) $R^2$ statistics between meQTLs and (B) $R^2$ statistics between meQTLs and established SS-associated SNPs at the MHC, for Europeans [7, 8].
(XLSX)

**S1 Text.**
(DOCX)

## Author Contributions

**Conceptualization:** Lindsey A. Criswell.

**Data curation:** Kimberly E. Taylor, Hong Quach, Diana Quach.

**Formal analysis:** Calvin Chi.

**Funding acquisition:** Lindsey A. Criswell.

**Investigation:** Lindsey A. Criswell, Lisa F. Barcellos.

**Methodology:** Calvin Chi.

**Project administration:** Hong Quach, Diana Quach.

**Resources:** Kimberly E. Taylor.

**Supervision:** Lindsey A. Criswell, Lisa F. Barcellos.

**Visualization:** Calvin Chi.

**Writing – original draft:** Calvin Chi.

**Writing – review & editing:** Kimberly E. Taylor, Lindsey A. Criswell, Lisa F. Barcellos.

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
