## [Decision Letter · Decision Letter 0]

1 Dec 2020

PONE-D-20-29702

Hypomethylation mediates genetic association with the major histocompatibility complex genes in Sjögren's syndrome

PLOS ONE

Dear Dr. Chi ,

Thank you for submitting your manuscript to PLOS ONE. After careful consideration, we feel that it has merit but does not fully meet PLOS ONE’s publication criteria as it currently stands. Therefore, we invite you to submit a revised version of the manuscript that addresses the points raised during the review process.

In particular, the reviewers raised several concerns about methodology and manuscript presentation, that I suggest to carefully address.

Further, please make all the data available. PLOS Data policy requires authors to make all data underlying the findings described in their manuscript fully available without restriction, with rare exception (please refer to the Data Availability Statement in the manuscript PDF file). The data should be provided as part of the manuscript or its supporting information, or deposited to a public repository. For example, in addition to summary statistics, the data points behind means, medians and variance measures should be available. If there are restrictions on publicly sharing data—e.g. participant privacy or use of data from a third party—those must be specified.

We look forward to receiving your revised manuscript.

Kind regards,

Annalisa Di Ruscio

Academic Editor

PLOS ONE

Journal Requirements:

Reviewers' comments:

Reviewer's Responses to Questions

**Comments to the Author**

1. Is the manuscript technically sound, and do the data support the conclusions?

Reviewer #1: Yes

Reviewer #2: Partly

2. Has the statistical analysis been performed appropriately and rigorously? 

Reviewer #1: Yes

Reviewer #2: Yes

3. Have the authors made all data underlying the findings in their manuscript fully available?

Reviewer #1: No

Reviewer #2: No

4. Is the manuscript presented in an intelligible fashion and written in standard English?

Reviewer #1: Yes

Reviewer #2: Yes

5. Review Comments to the Author

Reviewer #1: Chi et al report a control-case methylation study hinging on genetic association with MHC genes study in Sjögren's Syndrome (SS). SS is an autommune disease for which previous GWAS studies have shown association with polymorphism at the MHC genes. Previous research into the molecular etiology of this disease at the epigenetic level includes DNA methylation assays, certain of which revealed hypomethylation in the MHC loci (e.g. Imgenberg-Kreuz et al. 2016, ref. 21). Here, the authors report findings delving into the significance of the association between hypomethylation and genetic polymorphism, mostly focusing on the MHC locus. To date, conclusive association between the epigenetic regulation of MHC expression (which itself remains poorly understood) and polymorphism remains largely elusive. However, while the study is thus of potential interest for the field as it relies on labial salivary gland (LSG) biopsies (the standard diagnostic tool for SS as those contain tremendous amounts of B- and T-cells), it falls somewhat short of shedding definite light on the subject. A major critic for the study is the unclear status (and use, for a substantial part of the paper) of control group in the analyses (major points 1 and 2). Nevertheless, the study remains original; a plus of the study notably includes adapting the causal inference test (CIT) from Millstein et al. written to link eQTLs and gene expression. The analytic methods per se are sound and adequately described, the level of detail pertaining to the use of algorithms such as Combat, bumphunter as well as of pre- and post-processing of the data etc is appreciated. The large size of patient cohorts is also a strong point for the paper. That said, the manuscript also regrettably suffers from presentation issues which results in difficulties reading the manuscript, in particular with respect to the order and referencing of figure, with some panels either not referenced or referenced in a non-intuitive order, and some supplementary figures and tables are swapped; see minor point 1 for more ample details. There are some further conceptual details, notably re methodology and significance that also need addressing (minor point 1). Thus, the manuscript could use some major revamping from its current state which likely is not well suited for publication.

Major points:

1) While the first part of the results and Figure 2 tentatively validate the use of symptomatic non-cases as controls, their use is problematic particularly given the lack of information about possible secondary SS in patient diagnosis. Ideally, there should be an age- and ethnicity-matched healthy control group. However, the limited availability of clinical material, especially given the patient cohort sizes in the present study might understandably prevent that. At the very least, information about possible secondary SS should be included or samples with confirmed or suspected SLE or RA should be excluded if this is not possible or available. Case and non-case statistics (e.g. as in Cole et al. 2016 Table 1, reference 12) would also be helpful in this case.

2) Unless I missed it, the last part of the results (page 15, line 290 onwards, “DNA methylation mediates the effect of meQTLs on SS at the MHC”) does not include the use of non-case/control groups. Is it possible to run the CIT algorithm on meQTLs associated to non-case specific DMRs or DMR-SNP candidate pairs on the MHC locus, or to perform an analogous analysis entailing the use of non-cases as a negative control?

3) The statement about overlaps of DMRs from Cole et al. 2016 and the present study (page 12, line 265) should be accompanied with a hypergeometric test. Additionnally, the last part of the results (page 15) might benefit from hypergeometric tests to further highlight significance, if adequate.

4) Have the authors accounted for SNP density, which is higher at MHC genes and could result in DMRs being associated to SNPs by chance? Does selecting SNPs in SNP-dense regions at random result in similar results?

Minor points:

1) The presentation needs to be addressed. In particular, Figure 1 is not described in the text (it should be either in the methods or results), Fig 3C is referenced before 3A in different paragraphs, Fig 4C is referenced before 4A, Fig 4B is not referenced; Figs S1 and S3 are swapped, Figs S2 and S4 are swapped; Tables S2 and S5 are swapped, Tables S3 and S4 are swapped. The quality of the figures should also be worked on

2) It is unclear in the current manuscript what conclusions are to be drawn from the included ancestry information (Fig S2). This should either be expanded or removed altogether.

3) The submitted data does not seem to be readily available

4) Can the authors comment on the choice of a 250 kb window rather than 50 kb for SNPs from Smith et al 2014 (ref 37)?

5) There should be a sentence about DMR identification in the first part of the results (Page 11, line 232-233)

6) In Fig S2, what are C1 and C2?

Reviewer #2: Chi et al report a study of DNA methylation derived from labial salivary glands (LSGs) of Sjögren's syndrome (SS) and subsequent mediation analysis leveraging genetic data. They report overall 19 DMR-meQTL pairs, out of 215 observed DMRs. About half of these reside within the MHC and some are overlapping previously reported GWAS SS. The authors conclude that the SS MHC GWAS hits mediate their effect via hypomethylation. This is an interesting study with many merits, but some further clarifications are needed.

Results:

- What is the purpose of the 1st paragraph? In line 233 the authors refences DMRs that are introduced and estimated in the next section. The sentence “We observed that

233 CpG sites in DMRs significantly contributed to PC1 on average, with an average absolute 234 loading percentile of 94%” is not of importance. On the contrary, if one observes skewness of the PC loadings it is usually an indication of non-normal behavior of the PCs. Generally, the overall premise of the paragraph/PCA analysis is distracting. The objective of this paper is to identify DMRs and any possible mediation of the SS genetics (? See comment below). What the average reader expects in this first paragraph is an introduction of the cohort and the data.

- MHC DMRs: the authors did remove probes that overlapped polymorphic positions as part of their QC. This step is usually accomplished leveraging lists provided by various tools, e.g. minfi. Did the authors examined post-hoc whether any of the identified DMRs overlapped with any known variant, especially within the MHC?

- What is the justification of testing for meQTLs within =-250Kb and not a large region, e.g. +-1MBps? Given the long-range LD within MHC, one would expect this region to be larger for the MHC DMRs.

- The authors’ main finding, the one that dictates the title of the paper, is compressed in the last paragraph. It is not easy to easy to identify which are the six MHC variants that are reported in the SS GWAS and what were the reported ORs and p-values. For example, were these associations with the variants or respective HLA alleles? What is the LD of the reported MDR-meQTL with the GWAs hits? This part of the paper comes across as hastily put together although there is room to dig deeper into the reported associations.

- PCA plots: There are generally two PCA plots presented, Figure 2 and Sup Fig 1. Why do these PCA plots look so different? One would expect theses to be identical, given that the same exact data are utilized or should be utilized.

Figure 1: this is a simple representation of the genetic to methylation to phenotype model. It lacks other possible explanations of the causal relationship, e.g. reverse causation or independent associations. It is of little to no value and it should be removed.

Figure 2: how many probes were used for the PCA analysis? What do the authors mean by “preprocessed”? Do they imply QC-ed? This plot has a better place in the Supplementary Material rather than the main manuscript.

Figure 3: Panel B, could you replace the scientific notation with numbers? Especially the X axis can be represented in Mbps. Panel C, what is conveyed in this plot, especially in the X axis? What is the overall purpose of it? The main things one should review in the PC loadings are i) their distribution, and ii) the top loadings if the distribution is normal.

Figure 4: Panel A is of extremely low quality and there is not text visible. It cannot be evaluated what is plotted. Panel B is for which probe(s)/DMR? The respective legend does not explain which probe(s) is/are displayed. Panel C, is significance defined at p-value of <=0.05 as the legend suggests? The authors have not discussed the clear bimodal distribution of the “Significant” distribution. What is a possible explanation?

Supplementary Tables: Most seem to be mislabeled, e.g. Sup Table 4 is actually Sup. Table 5. More information is needed to explain what is displayed in each of the tables.

6. PLOS authors have the option to publish the peer review history of their article (what does this mean?). If published, this will include your full peer review and any attached files.

Reviewer #1: No

Reviewer #2: No

---

## [Author Response · Author response to Decision Letter 0]

9 Feb 2021

The reviewer and editor comments can be found in the cover letter and response letter as attached files.

---

## [Decision Letter · Decision Letter 1]

26 Feb 2021

Hypomethylation mediates genetic association with the major histocompatibility complex genes in Sjögren's syndrome

PONE-D-20-29702R1

Dear Dr. Chi,

We’re pleased to inform you that your manuscript has been judged scientifically suitable for publication and will be formally accepted for publication once it meets all outstanding technical requirements.

Kind regards,

Annalisa Di Ruscio

Academic Editor

PLOS ONE

Additional Editor Comments (optional):

Please confirm S3 Table is attached in the final version.

Reviewers' comments:

Reviewer's Responses to Questions

**Comments to the Author**

1. If the authors have adequately addressed your comments raised in a previous round of review and you feel that this manuscript is now acceptable for publication, you may indicate that here to bypass the “Comments to the Author” section, enter your conflict of interest statement in the “Confidential to Editor” section, and submit your "Accept" recommendation.

Reviewer #1: All comments have been addressed

Reviewer #2: All comments have been addressed

2. Is the manuscript technically sound, and do the data support the conclusions?

Reviewer #1: Yes

Reviewer #2: Yes

3. Has the statistical analysis been performed appropriately and rigorously? 

Reviewer #1: Yes

Reviewer #2: Yes

4. Have the authors made all data underlying the findings in their manuscript fully available?

Reviewer #1: Yes

Reviewer #2: Yes

5. Is the manuscript presented in an intelligible fashion and written in standard English?

Reviewer #1: Yes

Reviewer #2: Yes

6. Review Comments to the Author

Reviewer #1: All my concerns have been adequately concerned, I believe the manuscript has been substantially strengthened and now more clearly shows a link between between hypomethylation and genetic polymorphism at the MHC locus. Regarding the results included in the rebuttal letter, these are clear and have helped make a more compelling argument, particularly re the SNP density at the MHC locus; it is up to the authors to decide whether to include figures for those in the new supplementary results section. The manuscript reads well and the structure is clear; the presentation has very clearly improved.

Reviewer #2: The authors have adequately addressed all issues raised by the reviewers.

Minor comments:

- there is no S3 Table attached.

7. PLOS authors have the option to publish the peer review history of their article (what does this mean?). If published, this will include your full peer review and any attached files.

Reviewer #1: No

Reviewer #2: No

---

## [Editor Report · Acceptance letter]

13 Apr 2021

PONE-D-20-29702R1 

Hypomethylation mediates genetic association with the major histocompatibility complex genes in Sjögren's syndrome 

Dear Dr. Chi:

I'm pleased to inform you that your manuscript has been deemed suitable for publication in PLOS ONE. Congratulations! Your manuscript is now with our production department. 

Kind regards, 

on behalf of

Dr. Annalisa Di Ruscio 

Academic Editor

PLOS ONE